# Do-It-Yourself Preoperative High-Resolution Ultrasound-Guided Flap Design of the Superficial Circumflex Iliac Artery Perforator Flap (SCIP)

**DOI:** 10.3390/jcm10112427

**Published:** 2021-05-30

**Authors:** Daniel Schiltz, Jasmin Lenhard, Silvan Klein, Alexandra Anker, Daniel Lonic, Paul I. Heidekrueger, Lukas Prantl, Ernst-Michael Jung, Natascha Platz Batista Da Silva, Andreas Kehrer

**Affiliations:** 1Department of Plastic and Aesthetic Surgery, Hand Surgery, Helios Hospital Emil von Behring, 14165 Berlin, Germany; Daniel.Schiltz@helios-gesundheit.de; 2Department of Plastic, Hand and Reconstructive Surgery, University Hospital Regensburg, 93053 Regensburg, Germany; jasmin.lenhard@web.de (J.L.); silvan.klein@ukr.de (S.K.); alexandra.anker@ukr.de (A.A.); daniel.lonic@helios-gesundheit.de (D.L.); paul.heidekrueger@ukr.de (P.I.H.); lukas.prantl@ukr.de (L.P.); 3Department of Radiology, University Medical Center Regensburg, 93053 Regensburg, Germany; ernst-michael.jung@ukr.de (E.-M.J.); natascha.platz-batista-da-silva@ukr.de (N.P.B.D.S.)

**Keywords:** SCIP, ultrasonic flap design, microsurgery, CCDS, superficial circumflex iliac artery perforator flap

## Abstract

The superficial circumflex iliac artery perforator (SCIP) flap is a well-documented, thin, free tissue flap with a minimal donor site morbidity, and has the potential to become the new method for resurfacing moderate-size skin defects. The aim of this study is to describe an easy, reliable, systematic, and standardized approach for preoperative SCIP flap design and perforator characterization, using color-coded duplex sonography (CCDS). A list of customized settings and a straightforward algorithm are presented, which are easily applied by an operator with minimal experience. Specific settings for SCIP flap perforator evaluation were investigated and tested on 12 patients. Deep and superficial superficial circumflex iliac artery (SCIA) branches, along with their corresponding perforators and cutaneous veins, were marked individually with a permanent marker and the anatomy was verified intraoperatively. From this, a simplified procedure for preoperative flap design of the SCIP flap was developed. Branches could be localized and evaluated in all patients. A preoperative structured procedure for ultrasonically guided flap design of the SCIP flap is described. A 100% correlation between the number and emergence points of the branches detected by preoperative CCDS mapping and the intraoperative anatomy was found.

## 1. Introduction

In recent decades, the aims of reconstructive microsurgery have moved from the purely functional to the aesthetic realms, and did so while lowering donor site morbidity. Donor site cosmesis and concealment, limited scar lengths, tissue replacement following the “like-with-like” principle with thin, gliding, and pliable tissue properties, as well as many others aesthetic factors have become ever more important in reconstructive microsurgery. Respecting the angiosome concept in flap choice, perforator flaps are increasingly required [1,2,3,4]. The controlled thickness of free flaps is crucial for the cosmesis of the donor site and the defect area. In addition, the harvest of thinner perforator flaps in the superficial fascial plane is described in the literature [5]. The superficial circumflex iliac artery perforator (SCIP) flap is a well-described [5,6,7,8] thin, free tissue flap that offers minimal donor site morbidity, the possibility of chimeric design, and the option to include lymph nodes [9] in the recipient area. The SCIP flap has the potential to be applied for resurfacing moderate-size skin defects [10,11,12,13]. 

Preoperative flap design and planning are essential for the success of microsurgical operations, especially for inexperienced surgeons. However, the anatomy of the SCIP flap site is described as quite complex and variable [14,15]; a recent anatomical study showed a reliable anatomy of the deep and superficial branches of the SCIA [13]. Anatomical features of the flap are a rather short pedicle length (superficial branch: +/−6.6 cm; deep branch: +/−9.1 cm [13]) with a small vessel diameter, limited flap width if closure is intended, and the variability in the perforator that supplies the flap. It was shown in 142 patients [4], using computed tomographic angiography and surgical anatomy, that the origin of the superficial branch of the superficial circumflex iliac artery (SCIA) is always seen to penetrate the deep fascia. These findings were recently confirmed on 21 cadavers [13]. However, the pathway of the superficial branch can either anchor directly into the dermis or extend as an axial pattern artery, implying a varying chance of survival of the flap [4]. Contrary to its nomenclature, the SCIP flap is in reality an axial flap with a superficial and a deep branch of the SCIA giving off multiple small branches to the skin paddle (Figure 1 and Figure 2). Preoperative flap planning with computed tomography angiography (CTA) is a safe option [16,17]; however, it does imply a radiation burden and the need for involvement of a radiology department. A harmless and simple alternative is the color Doppler ultrasound assessment [18,19,20,21]. Color-coded duplex sonography (CCDS) has been proven to be a powerful tool for preoperative perforator characterization when using a structured approach and mapping algorithm [22,23]. For perforator mapping in preoperative flap planning of the anterolateral thigh (ALT), color Doppler ultrasonography was shown to have the highest pooled sensitivity and positive predictive value compared with a handheld Doppler [24,25]. This method uses the Doppler effect and overlays flow information (in color on a grayscale background) in a real-time image, displaying the different velocities and directions of vascular blood flow [24]. SCIP flap design may thus be simplified using CCDS. The aim of this study is to construct a reliable, systematic and standardized approach for an easy preoperative SCIP flap design as well as perforator characterization through CCDS. The approach should not require in-depth knowledge or expertise of ultrasound technology, and therefore be usable by the majority of microsurgeons. 

## 2. Methods

This study was planned as a retrospective cohort study. Patients who had a soft tissue defect reconstruction using a free SCIP flap transfer carried out between January 2018 to October 2020 were included in the study, and had their preoperative flap design planning and perforator characterization performed using CCDS. All patients were treated at the University Hospital Regensburg. The study was approved by the Ethics Committee of the University of Regensburg (reference number: 18-1133-101) and was planned in accordance with the Helsinki Declaration of 1975. All flaps were operated on and designed with CCDS by the senior author (A.K.).

### 2.1. Medical Indication for SCIP Flap

There were no limitations on the localization of the soft tissue defect. The indication for the SCIP flap coverage was the need for a thin perforator flap, a small or midsize defect and the absence of previous operations in the groin. 

### 2.2. Ultrasound-Based Flap Design & Perforator Characterization

US flap planning was performed by a surgeon (A.K.) the day before surgery and reconfirmed on the operating table before dissection. Different linear multifrequency transducers, including a 12L-RS (5–13 MHz), a 9L (6–9 MHz), and a ML6-15-D (4–15 MHz) with a LOGIQ E9 US device (GE Healthcare, Milwaukee, WI, USA) were used. Qualitative and quantitative perforator evaluation was performed using the method described recently by our research team [26]. For each patient, optimal settings for SCIP flap perforator evaluation were assessed. These settings were then compared and are presented in a list (Table 1). A simplified procedure for preoperative perforator mapping and characterization was then developed. After localization of the superficial and deep branches and the emergence point where the branches perforate the deep and superficial fascia, these positions (perforators) were marked with a permanent marker (Figure 3). The flap was then designed according to the perforator location and with respect to the defect size, so that a primary closure of the groin region was guaranteed (pinch test). 

### 2.3. Operation Technique

The harvest of the SCIP flap was performed from the latero-cranial to the medio-caudal areas as described previously [27,28]. The SCIP flaps were harvested from lateral along the superficial fascia up to the point of emergence of the perforator. Subsequently, the perforator was prepared to the SCIA. The dissection was conducted at the level of the superficial fascia. After a medial perforator was found, dissection continued deep following the superficial branch to completely expose the superficial circumflex iliac artery at its exit from the femoral artery. In the case of an absence of a sizable associate vein, a superficial cutaneous vein was chosen and included in the flap design. After the flap was harvested, microsurgery was performed with an operative microscope (Kinevo, Carl Zeiss Meditec AG, Germany). Arterial anastomoses were performed using 8-0 or 9-0 sutures (Ethilon, Ethicon inc, Somerville, NJ, USA). Venous anastomosis was performed with a coupler device (Synovis Micro Companies Alliance, Inc, Birmingham, AL, USA).

## 3. Results

There was a total of 12 patients included in the study. The mean age was 42.9 years, the mean body mass index (BMI) was 26.77 kg/m^2^ (range: 18.65–40.91 kg/m^2^), and the male to female ratio was 8:4. An overview of patient data is shown in Table 2. 

All patients underwent a high-resolution ultrasound-guided flap design as described below. SCIA, branches, and perforators could be localized and evaluated in all patients. A preoperative structured procedure for ultrasound-guided flap design of the SCIP flap was then developed (Figure 4). We recommend the following approach: 

First, a qualitative characterization of perforator(s) should be performed. This means the examiner should visualize tissue morphology to obtain an overview of the anatomy, localize the SCIA and their branches as well as emergence points of perforators. In “B-mode” (brightness mode), static anatomy can be visualized. Gain should not be set too high for better contrast. In “Color Flow” (CF) mode, blood flow can be visualized and the identified vessels can be confirmed.

▪Start in B-mode. Choose a linear transducer (9–15 MHz). In this study, the high-resolution transducer ML6-15-D (4–15 MHz) demonstrated the highest level of detail for displaying perforators. US devices offer presets that can be chosen before imaging. The “Thyroid” preset is recommended, as these settings are close to those optimized for the SCIP flap (Table 1). Using transverse cuts, the examiner should visualize the anatomy, assess tissue morphology, and localize the SCIA branching off the femoral artery (FA). ▪Color Flow (CF)-mode should then be initiated. This mode allows for color-coded imaging of the intravascular blood flow. The “PRF/Scale” should be moved between 0.7 and 1.3 kHz/5 and 9 cm/s to be able to visualize smaller vessels with low flow (perforators, superficial arteries and veins). By micro-rotation and careful sliding of the transducer, the examiner should verify the anatomy, asses the vascular axis of the superficial branch of SCIA, and confirm the positions of one or several perforators branching off the superficial branch. The emergence points of the perforator penetrating the deep as well as the superficial fascia should be marked with a permanent marker to allow for a precise flap design and to simplify microsurgical dissection. An additional cutaneous vein may be marked and traced deep to the femoral vein at the level of the saphenous hiatus. 

After qualitative characterization, quantitative characterization may be performed to gain information about the physiological parameters of perfusion as well as vessel diameter and blood flow velocities. We suggest these parameters be measured at the emergence points of the perforator penetrating the deep fascia. It allows a comparison with the values of other perforators, so that the perforator choice can be taken before surgery. The diameter can additionally be measured distally, where the SCIA branches off the femoral artery at the level that microsurgical anastomosis will be performed. Diameters may be measured in CF-mode with the “measurement” tool. Blood flow velocities may be quantified using “Pulse Wave (PW)-Mode”. In this mode, the pulse systolic velocity (PS) and end diastolic velocity (ED) as well as the resistance index (RI) may be measured. Vascular diameter and blood flow velocities may then serve as an orientation to compare the different perforators (Figure 5).

Following this approach, flap design was performed as described in the literature [10,27,28] and vessels were marked with a permanent marker on the patient. A 100% correlation between the number and emergence points of the perforators detected by pre-operative CCDS mapping and intraoperative anatomy was found. Table 3 shows the results of CCDS and intraoperative anatomical findings of the flaps. The mean perforator diameter measured by CCDS was 1.54 ± 0.38 mm. Mean PS was 16.82 ± 4.19 cm/s. The mean ED was 4.61 ± 1.31 cm/s and the mean resistance index (RI) was 0.69 ± 0.09.

In all cases, the SCIP flaps were based on a single perforator originating from the medial superficial branch of the SCIA. All harvesting sites could be closed primarily and there were no complications noted at the donor sites. One surgical revision of the arterial anastomosis had to be performed. In total, there were two total flap losses that were not attributed to the methodology of the SCIP flap design by CCDS but poor recipient vessel quality. No partial necrosis occurred. 

## 4. Discussion

The SCIP flap has the potential to become the new workhorse for resurfacing moderate-size skin defects [10,11,12]. The anatomy is still inconsistently discussed in the literature and is described as variable to a certain degree [14,15,16]. A recent anatomical study of 21 cadavers brought more clarity to the anatomy of the SCIP and found the anatomy to be reliable [13]. Thus, preoperative flap planning and design are important. CTA has proven to be a reliable tool for general perforator mapping in preoperative perforator flap planning [29,30]. Recently, a CTA-based planning method was published by Pereira et al. [17]. The disadvantages of CTA and digital subtraction angiography (DSA) are the use of a contrast medium and radiation exposure to the patient. Patients with renal insufficiency or metal implants may be not be candidates for such modalities. Furthermore, CTA implies the dependence on a radiology department. CCDS is a far more cost-effective and easier alternative for perforator mapping of the SCIP flap [18,19,20,21]. It may be performed independently from a radiology department by a microsurgeon. In a recent study, Yoshimatsu et al. compared hand-held Doppler with CCDS as preoperative flap planning tools, and found that preoperative high-resolution ultrasound systems significantly decreased the intraoperative conversion rate of the SCIP flap pedicle from the superficial branch to the deep branch of the SCIA, thus notably facilitating the procedure [31]. 

As CCDS devices are mobile, perforators can be reevaluated at any time without a radiation burden. The do-it-yourself preoperative high-resolution ultrasound-guided flap design of the SCIP flap is an easy and fast method. The novelty of this study is that it provides a structured methodology for SCIP flap micro-vessel characterization, using CCDS including precise device settings. Ultrasound devices nowadays are omnipresent in hospitals. We would like to encourage microsurgeons to start mapping perforators and comparing their preoperative and intraoperative anatomical findings. One primary goal of our work is therefore to make the SCIP flap accessible to a larger number of microsurgeons. Performing preoperative micro-vessel mapping should simplify the understanding of the challenging surgical anatomy of SCIP flaps and enhance surgical planning, e.g., detection of an additional vein for improved flap drainage. This should improve the confidence of more microsurgeons to harvest SCIP flaps, and increase patient safety as the vascular anatomy is preoperatively marked on the skin for enhanced intraoperative orientation. Device calibration with adequate settings through our guide with just little practice should even enable beginners of CCDS and less-experienced microsurgeons to successfully achieve a level of expertise where what you see is what you get (intraoperatively). Further, micro-vessel characterization by determined by measuring quantitative parameters such as vascular diameter and blood flow velocities may possibly be useful for matching the flow velocity with the recipient vessels, e.g., if perforator-to-perforator anastomoses are used as described by Suh et al. [2], it also allows for the comparison of values of SCIP micro-vessels with those of different (perforator) flaps [22,32,33,34]. Perforators of other flaps are usually also measured at their emergence point at the level of the deep fascia. For the future, knowledge of flow velocities and vascular diameters in comparison with the size of the nourished angiosomes may help to gain a better estimation of how to design flap perimeters in larger flaps without vascular compromise. Undeniably, more research of these parameters and their specific effects on flap survival, possible flap size, and thickness is needed. Therefore, we would like to encourage other groups to measure these (and more) quantitative parameters in different flaps. Overall, this will ultimately lead to a much better understanding of the intricate perforator flap physiology.

Remarkably fast detection times for ALT perforators with a median CCDS assessment duration of only 3 min along with a high level of precision was demonstrated in the study by Debelmas et al. with evidence at level II [35]. Furthermore, a 100% correlation between the number and emergence positions of the perforators detected by preoperative CCDS mapping and intraoperative anatomy was found in our study. This shows the high reliability of CCDS perforator mapping. 

CCDS provides quantitative (number, localization) as well as qualitative (static, dynamic) information about perforators [22,36,37]. An important quantitative static parameter is the perforator diameter at the emergence point of the perforator penetrating the deep fascia. The ideal skin perforator for perforator flaps is defined as one larger than 0.5 mm with an adequate waveform of pulsatile blood flow [38,39]. The mean diameter measured by CCDS (1.54 ± 0.38 mm) of the SCIP flap is comparable to that of the profunda artery perforator (PAP) flap (1.496 ± 0.39 mm), which was assessed by our research team in previous research [23]. Berner et al. found the median diameter of the perforator and its concomitant vein was 1 mm (range 0.8–2 mm) [7]. Although contrary to its nomenclature, the SCIP flap may in reality be an axial flap with both the superficial or the deep branch of the SCIA giving off multiple small branches (perforators) to the skin. The perforator diameter of the SCIP flap seems to be fairly constant. However, the diameters measured by CCDS could not be confirmed intraoperatively in this study as the assessment of the diameter with a ruler was considered inaccurate. 

In a previous study, using the augmented power-Doppler (PD)-mode in CCDS, the perforator artery size was overestimated in 65.5% of the cases by 50% or more compared with the diameter observed clinically in a series of 12 PAP flaps [23]. Size mismatches of radiologically measured perforator diameters compared with clinical findings also occurred in other modalities such as CTA and MRA [40]. It was demonstrated that size estimations based on cross-sectional imaging (CTA/MRA) are even less accurate than those based on CCDS [41]. Diameter may additionally be measured more distally, where the SCIA branches off the FA and the microsurgical anastomosis will be performed. This might ensure the surgeon of a sufficient vessel diameter for anastomosis. In addition to providing static qualitative and quantitative parameters such as the number of perforators, localization and diameter, CCDS is also capable of providing dynamic qualitative information. Hemodynamics, flow direction, and muscular/subcutaneous perforator route can be assessed and visualized as well [36,37]. 

The PS, ED, and RI were also assessed. Recent findings suggest that when planning perforator-to-perforator anastomosis in flaps, matching the flow velocity of the pedicle with the recipient vessels may be beneficial [2]. A PSV of 15 cm/s and higher was recommended as a minimum value for the perforator selection of the planned flap. The mean PS of the SCIP flap was higher than this value (16.82 ± 4.19 cm/s) in this study. The recipient artery may be suitable if the PSV is greater than 15–20 cm/s [2]. 

All data presented in this study were assessed with a ML6-15-D (4–15 MHz) transducer and a LOGIQ E9 US device. Other transducers and devices were tested and it was found that the precise settings varied slightly, although with most of the devices and transducers, the ideal settings were close to the range of values presented in Table 1. 

A clear limitation of this study is that the anatomical data were not completely assessed intraoperatively. In particular, the pedicle length, diameter, and point of perforation of the deep fascia were not assessed. Furthermore, only data of the superficial branch were assessed. As in all cases, the need of a thin free flap was the indication for SCIP flap and harvesting the SCIP flap including the superficial branch allows a much thinner harvest [13]. In none of the cases was a switch to the deep branch necessary. Another advantage of CCDS, which was not analyzed in this study, might be the possible identification of other perforators, allowing a superficial plane only raising of the SCIP flap.

All patients were monitored postoperatively at our ward for plastic surgery. Trained nurses performed clinical flap monitoring and evaluation as previously published by our research team [31,42]. In a review of 210 SCIP flaps, 10% underwent revision surgery. Eight of these were caused by arterial thrombosis, four were caused by venous thrombosis, three were caused by a combination of arterial and venous thrombosis, four were caused by hematoma formation, and two were negative explorations. Six flaps (2.9%) were lost despite revision [10]. Another series of 20 free SCIP flap transfers described a 100% flap survival [7].

## 5. Conclusions

Color-coded duplex sonography is a reliable and easy tool which can be used to make the planning of preoperative flap design and perforator mapping of the SCIP flap safer. A 100% correlation rate between the number and emergence positions of the perforators detected by preoperative CCDS mapping and intraoperative anatomy was found. 

## Figures and Tables

**Figure 1 jcm-10-02427-f001:**
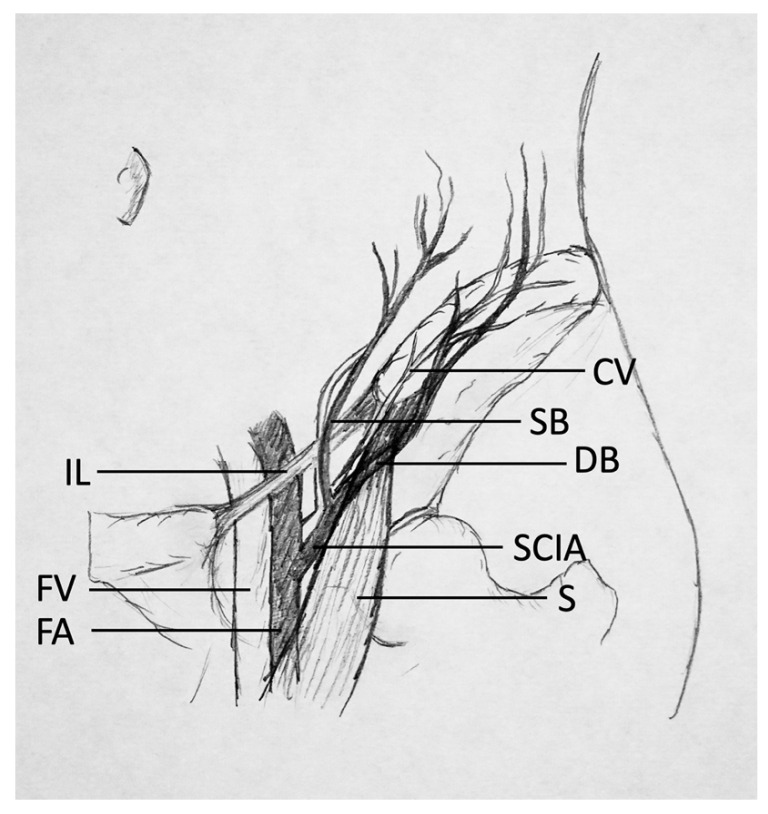
Simplified overview of the anatomy of the groin. IL = inguinal ligament; FV = femoral vein; FA = femoral artery; CV = cutaneous vein; SB = superficial medial branch of SCIA; DB = deep lateral branch of SCIA; SCIA = superficial circumflex iliac artery; S = sartorius.

**Figure 2 jcm-10-02427-f002:**
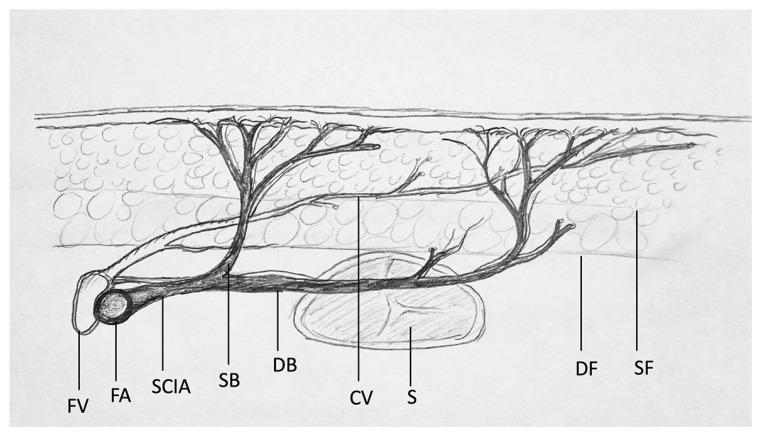
Simplified overview of the anatomy of the SCIA. FV = femoral vein; FA = femoral artery; CV = cutaneous vein; SB = superficial medial branch of SCIA; DB = deep lateral branch of SCIA; SCIA = superficial circumflex iliac artery; S = sartorius; DF = deep fascia; SF = superficial fascia. Quantitative parameters such as vascular diameter and blood flow properties such as pulse systolic velocity (PS), end diastolic velocity (ED), as well as the resistance index (RI) were measured at the emergence point of vessels at the level of penetration of the deep fascia.

**Figure 3 jcm-10-02427-f003:**
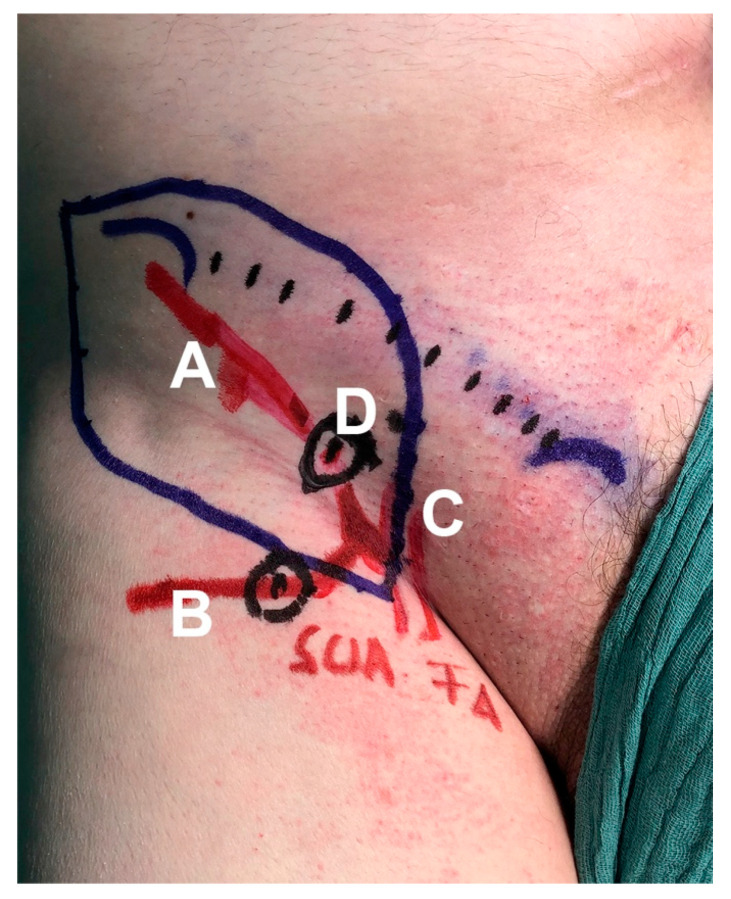
After localization of the SCIA branches (A = medial superficial branch, B = lateral deep branch), the femoral artery (C) and the emergence point, where the medial superficial branch perforates the deep and superficial fascia (D); these findings were marked individually with a pen.

**Figure 4 jcm-10-02427-f004:**
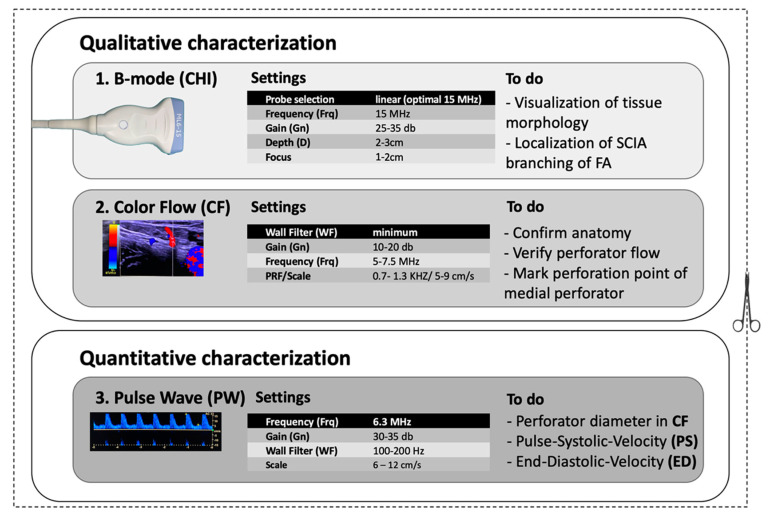
Structured procedure for ultrasound-guided flap design of the SCIP flap.

**Figure 5 jcm-10-02427-f005:**
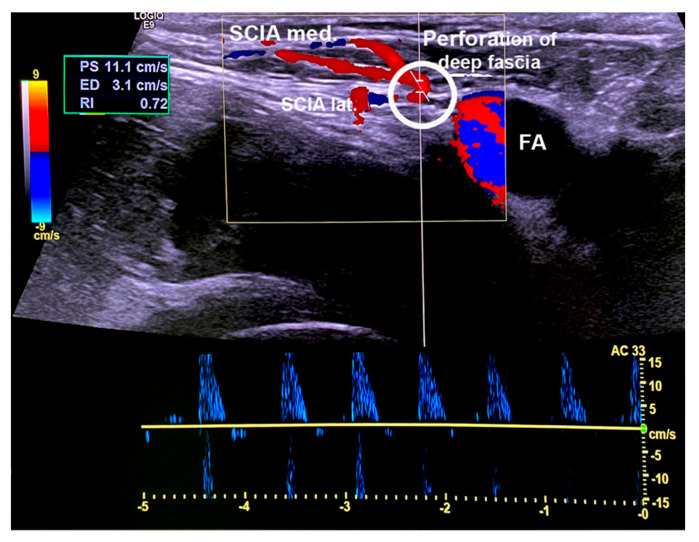
Screen shot in Pulse Wave (PW) mode while measuring pulse systolic velocity (PS), end diastolic velocity (ED), and the resistance index (RI) at the emergence point of the medial superficial branch perforating the deep fascia (SCIA med.). In the same plane, the lateral deep branch of the SCIA (SCIA lat.) can be visualized. FA = femoral artery.

**Table 1 jcm-10-02427-t001:** Recommended presets for perforator evaluation of the SCIP flap.

Qualitative Characterization	
**B-mode (B)**	
Probe selection	linear (optimal 15 MHz)
Frequency (Frq)	15 MHz
Gain (Gn)	25–35 dB
Depth (D)	2–3 cm
Focus	1–2 cm
**Color flow (CF)**	
Wall Filter (WF)	minimum
Gain (Gn)	10–20 dB
Frequency (Frq)	5–7.5 MHz
PRS/Scale	5–9 cm/s
**Quantitative Characterization**	
**Pulse Wave (PW)**	
Frequency (Frq)	6.3 MHz
Gain (Gn)	30–35 dB
Wall Filter (WF)	100–200 Hz
PRS/Scale	6–12 cm/s

**Table 2 jcm-10-02427-t002:** Overview of general patient data.

Patient No.	Sex	Age (years)	BMI (kg/m^2^)	Tissue Defect Area
1	F	47	29.64	ankle
2	M	58	40.91	forefoot
3	M	68	31.74	forefoot
4	F	17	23.34	ankle
5	M	43	28.29	ankle
6	M	56	26.34	forefoot
7	F	22	20.76	head
8	M	49	24.84	hand
9	M	9	24.97	forefoot
10	F	25	18.65	forefoot
11	M	58	26.43	ankle
12	M	55	25.36	forefoot

M = male; F = female.

**Table 3 jcm-10-02427-t003:** Flap and perforator characteristics.

	CCDS Characteristics					Intraoperative Characteristics		
	Qualitative Characteristics		Quantitative Characteristics					
Patient No.	Mapped Perforator	Perforator Diameter CCDS (mm)	PS cm/s	ED cm/s	RI	Intraoperatively Confirmed Anatomy	Flap Size (cm)	Recipient Vessel
1	medial	1.3	14.7	3.1	0.79	yes	15 ×6	anterior tibial artery
2	medial	1.7	24.5	6.8	0.72	yes	22 × 8	anterior tibial artery
3	medial	1.2	17.9	5.7	0.68	yes	16 × 7	anterior tibial artery
4	medial	2.3	16.4	2.1	0.87	yes	8 × 10	dorsalis pedis artery
5	medial	1.6	9.9	4.5	0.55	yes	20 × 7	posterior tibial artery
6	medial	1.2	17.9	5.7	0.68	yes	/	dorsalis pedis artery
7	medial	1.9	17.7	4.9	0.61	yes	4 × 5	superficial temporal artery
8	medial	1.2	20.6	5.2	0.58	yes	13 × 6	perforator of radial artery
9	medial	1.4	11	4.6	0.64	yes	15 × 4.5	dorsalis pedis artery
10	medial	1.3	16.3	5.3	0.68	yes	/	anterior tibial artery
11	medial	2.2	12.6	2.6	0.79	yes	11 × 18	posterior tibial artery
12	medial	1.2	22.3	4.9	0.78	yes	15 × 6	anterior tibial artery

PS = pulse systolic velocity; ED = end diastolic velocity; RI = resistance index; A = Arteria; / = missing data.

## Data Availability

Data available on request due to restrictions eg privacy or ethical. The data presented in this study are available on request from the corresponding author. The data are not publicly available due to privacy police.

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
