# Peer review of "Do-It-Yourself Preoperative High-Resolution Ultrasound-Guided Flap Design of the Superficial Circumflex Iliac Artery Perforator Flap (SCIP)"

_jcm, 2021, doi:10.3390/jcm10112427_

Round 1

Reviewer 1 Report

I would like to congratulate and thank the authors for the possibility of reviewing their work. I generally believe that the detailed methodology for the CCDS is of real benefit for microsurgeons looking to further increase their use of the SCIP flap and therefore would support the publication of this manuscript if some major and minor issues are resolved.  

Major issues:

  1. The publication rhythm regarding SCIP flaps is very rapid, and some newly published papers need to be addressed, discussed and cited. On the one hand, a recent anatomical study (Zubler et al. The anatomical reliability of the superficial circumflex iliac artery perforator (SCIP) flap. Ann Anat. 2021 Mar;234:151624.) provides distinct clarity for the anatomy of the different layers and SCIA perforators of the SCIP and should be cited and discussed in the parts related to inherent anatomical variability of the SCIP vascular anatomy. On the other hand, a recent study (Yoshimatsu H et al. Use of Preoperative High-Resolution Ultrasound System to Facilitate Elevation of the Superficial Circumflex Iliac Artery Perforator Flap. J Reconstr Microsurg. 2021 Apr 14.) is directly related to the authors manuscript. Its findings need to be discussed and evaluated especially regarding differences to the actual paper.
  2. While the description of the CCDS method says that perforators through the superficial fascia were analyzed for the different branches of the SCIA, the data presented resumes only results regarding the medial, main perforator of the superficial SCIA branch and only at the level of the deep fascial penetration. Information on other perforators and a discussion why always the superficial branch was chosen is required. The main medial perforator is easily found in experienced hands also with hand-held Doppler, the main advantage of high-resolution CCDS may lie in identifying other perforators, allowing a superficial plane only raising of the SCIP flap.
  3. No data is provided on the outcome of the flaps, see also minor comment lines 192-194. Since the method aims to make the SCIP flap safer it is extremely important to know if the flaps survived completely or had partial necrosis.

Minor:

Lines 124-126: It is unclear if the medialmost part of the SCIP flap was also raised on the superficial fascia or “en bloc” with the main vessel in a deep layer once the perforator was found. Intraoperative images of a flap example would be very useful.

Lines 129: Replace suture sizes to 8-0 and 9-0

Lines 192-194: Please also address complications at the recipient site. Did all flaps survive, was there any partial necrosis? Reconstructive success? While it is clear that there was one arterial revision, it is not stated if there were or not any other complications.

Lines 288-295: Here some data is cited regarding flap loss and flap revision rates in the literature while there is no reference to the outcomes of the flaps in this study.

Author Response

Point-to-point reply

to the manuscript no. jcm-1200352, entitled “Do-it-yourself preoperative high-resolution ultrasound guided flap design of the superficial circumflex iliac artery perforator flap (SCIP)”

Reviewer 1

I would like to congratulate and thank the authors for the possibility of reviewing their work. I generally believe that the detailed methodology for the CCDS is of real benefit for microsurgeons looking to further increase their use of the SCIP flap and therefore would support the publication of this manuscript if some major and minor issues are resolved.  

Dear reviewer, thank you very much for carefully reading our manuscript and we really appreciate your comments. In the following, please find our point-by-point replies to your comments and according changes in the manuscript. Changed parts in the manuscript are marked up using “track changes”.

Major issues:

  1. The publication rhythm regarding SCIP flaps is very rapid, and some newly published papers need to be addressed, discussed and cited. On the one hand, a recent anatomical study (Zubler et al. The anatomical reliability of the superficial circumflex iliac artery perforator (SCIP) flap. Ann Anat. 2021 Mar;234:151624.) provides distinct clarity for the anatomy of the different layers and SCIA perforators of the SCIP and should be cited and discussed in the parts related to inherent anatomical variability of the SCIP vascular anatomy.

Authors reply:

Thank you for citing this relevant recent anatomical study of the SCIP flap. We now cited the article and adjusted the introduction respectively. Furthermore, we discussed the anatomical findings in the discussion.

On the other hand, a recent study (Yoshimatsu H et al. Use of Preoperative High-Resolution Ultrasound System to Facilitate Elevation of the Superficial Circumflex Iliac Artery Perforator Flap. J Reconstr Microsurg. 2021 Apr 14.) is directly related to the authors manuscript. Its findings need to be discussed and evaluated especially regarding differences to the actual paper.

Authors reply:

We agree with the reviewer that this important paper is related to our manuscript. Although there are some major differences in focus of the respective studies. Yoshimatsu et al. analyzed the conversion rate from the superficial to the deep branch of the SCIA during the flap harvest in two patient groups. One cohort had preoperative flap planning through hand-held doppler and the other group had CCDS examination.  The secondary outcomes were the number of venous anastomoses (one or more than one), operative time (minutes), and flap complications. Even more important, there are major differences in the ultrasound technology applied. The group of Yoshimatsu are using a very expensive, super-high end ultrasound device system (Vevo MD, ultrasound device, Fujifilm Visual Sonics, Amsterdam, the Netherlands) with a stunningly high MHz transducer of 48 MHz that is currently only available in very few highly specialized microsurgical departments around the world specifically conducting clinical research in lymphatic surgery. Additionally, the ultrasound mode used in their work is regular B-Mode, but at a very high resolution. B-Mode in general has its domain in displaying tissue morphology, but does not provide any information about flow direction (value for distinguishing veins from arteries, especially in small microvessels) and flow velocity. 

 We refer to this important article in the discussion.

The novelty of our study is that it provides a structured methodology for not only for SCIP flap micro vessel detection, but also characterization applying Color-Coded Duplex Sonography (CCDS). CCDS features Color Flow (CF)-mode as well as Pulse Wave (PW) mode which are different modes from B-Mode. CF- mode is generally used to display blood flow, blood flow direction and measure vessel diameter. It is therefore ideal not only to detect but further assess and characterize blood vessels. PW-Mode adds even more information to further evaluate SCIP-Vessels as blood flow is quantified. Small arteries may be reliably distinguished from veins. In this mode, the “Pulse-Systolic-Velocity” (PS) and “End-Diastolic-Velocity” (ED) as well as the “Resistance-Index” (RI) may be measured. Vascular diameter and blood flow velocities may then serve as an orientation to compare the different perforators. Thus, different modes and a characterization of vascular anatomy for SCIP flaps are presented in our study. Reference values are provided for comparability.  Moreover, our paper describes precise device settings how to precisely trim ultra sound devices to make them applicable for the design of thin perforator flaps as the SCIP. Our paper is supposed to address microsurgeons interested in learning CCDS technology. The paper also provides a structured methodology that simplifies pre-operative markings and flap design in order to improve safety and efficiency. In comparison with Yoshimatsu’s work, the device applied in our study using a 15 MHz transducer is by far more comparable to any standard contemporary ultrasound machine available in almost any larger hospital. Thus, the target group of microsurgeons having devices available to them compared to the device used in our study is by far much larger than the ones with access to ultra-high-performance devices with transducers of 48 MHz such as it was used in their study.

  1. While the description of the CCDS method says that perforators through the superficial fascia were analyzed for the different branches of the SCIA, the data presented resumes only results regarding the medial, main perforator of the superficial SCIA branch and only at the level of the deep fascial penetration. Information on other perforators and a discussion why always the superficial branch was chosen is required. The main medial perforator is easily found in experienced hands also with hand-held Doppler, the main advantage of high-resolution CCDS may lie in identifying other perforators, allowing a superficial plane only raising of the SCIP flap.

Authors reply:

We agree with the reviewer. In this study, only the superficial branch was analyzed, as indication for SCIP flap was the need of a thin flap and ease of harvest. Therefore, the SCIP was always harvested with the superficial branch. When compared to a hand-held Doppler CCDS provides a clear visualization of the anatomy (both branches, veins, potential anatomical variances, course and depth of the vessels, lymph nodes to spare). Further advantages are, as the reviewer mentioned, the possible identification of additional perforators, enabling the surgeon to dissect the pedicle first and allowing a superficial plane only raising of the SCIP flap. We added this in the discussion.

  1. No data is provided on the outcome of the flaps, see also minor comment lines 192-194. Since the method aims to make the SCIP flap safer it is extremely important to know if the flaps survived completely or had partial necrosis.

Authors reply:

Thank you for your helpful comment. In all cases the SCIP flaps were based on a single perforator originating from the medial branch of the SCIA. All donor sites could be closed primarily and there were no complications noted at the donor sites. One surgical revision of the arterial anastomosis had to be performed. In total, there were 2 total flap losses which were not attributed to the methodology of the SCIP flap design by CCDS but poor recipient vessel quality.  We added this in the result section.

Minor:

Lines 124-126: It is unclear if the medial most part of the SCIP flap was also raised on the superficial fascia or “en bloc” with the main vessel in a deep layer once the perforator was found. Intraoperative images of a flap example would be very useful.

Authors reply:

Thank you for your helpful comment. The SCIP flaps were harvested from lateral along the superficial fascia up to the point of emergence of the perforator. Than the perforator was prepared to the SCIA. We clarified this in the manuscript.

Lines 129: Replace suture sizes to 8-0 and 9-0

Authors reply:

we apologize for the unclear wording and have changed the manuscript accordingly.

Lines 192-194: Please also address complications at the recipient site. Did all flaps survive, was there any partial necrosis? Reconstructive success? While it is clear that there was one arterial revision, it is not stated if there were or not any other complications.

Lines 288-295: Here some data is cited regarding flap loss and flap revision rates in the literature while there is no reference to the outcomes of the flaps in this study.

Authors reply:

we apologize for the unclear wording and have clarified this the manuscript. In total, there were 2 total flap losses, but were not attributed to the methodology of the SCIP flap design by CCDS but poor recipient vessel quality.   No partial necrosis occurred.

Reviewer 2 Report

The authors introduced the used of CCDS and applied as preoperative tool before SCIP elevation. The applied this technique in 12 patients.

Unfortunately no novelty in this report.

Recently Yoshimatsu published very similar paper: (actually more cases and compared as well handheld Doppler ultrasonography and HRUS)

Use of Preoperative High-Resolution Ultrasound System to Facilitate Elevation of the Superficial Circumflex Iliac Artery Perforator Flap.

Yoshimatsu H, Karakawa R, Fuse Y, Okada A, Hayashi A, Yano T.J Reconstr Microsurg. 2021 Apr 14. doi: 10.1055/s-0041-1726395. Online ahead of print.PMID: 33853132   I do not see much more informations in your paper.

Author Response

Point-to-point reply

to the manuscript no. jcm-1200352, entitled “Do-it-yourself preoperative high-resolution ultrasound guided flap design of the superficial circumflex iliac artery perforator flap (SCIP)”

Reviewer 2

The authors introduced the used of CCDS and applied as preoperative tool before SCIP elevation. The applied this technique in 12 patients.

Unfortunately no novelty in this report.

Recently Yoshimatsu published very similar paper: (actually more cases and compared as well handheld Doppler ultrasonography and HRUS)

Use of Preoperative High-Resolution Ultrasound System to Facilitate Elevation of the Superficial Circumflex Iliac Artery Perforator Flap.

Yoshimatsu H, Karakawa R, Fuse Y, Okada A, Hayashi A, Yano T.J Reconstr Microsurg. 2021 Apr 14. doi: 10.1055/s-0041-1726395. Online ahead of print. PMID: 33853132   I do not see much more informations in your paper. 

Authors reply:

Dear reviewer, thank you very much for carefully reading our manuscript and we appreciate your comments. We agree with the reviewer that this important paper is partially related to our manuscript. However, there are some major differences in focus of the respective studies. Yoshimatsu et al. analyzed the conversion rate from the superficial to the deep branch of the SCIA during the flap harvest in two patient groups. One cohort had preoperative flap planning through hand-held doppler and the other group had CCDS examination.  The secondary outcomes were the number of venous anastomoses (one or more than one), operative time (minutes), and flap complications. The novelty of our study is that it provides a structured methodology for SCIP flap micro vessel characterization using CCDS including precise device settings. If you compare any literature available, very little focus is placed on teaching the reader actually how to correctly fine-tune ultrasound devices to precisely detect and characterize SCIP and other perforators for flap design. The aim of our article is to bring CCDS closer to the reader and allow him/her to perform CCDS.

Even more important, there are major differences in the ultrasound technology applied. The group of Yoshimatsu are using a very expensive, super-high end ultrasound device system (Vevo MD, ultrasound device, Fujifilm Visual Sonics, Amsterdam, the Netherlands) with a stunningly high MHz transducer of 48 MHz that is currently only available in very few highly specialized microsurgical departments around the world specifically conducting clinical research in lymphatic surgery. Additionally, the ultrasound mode used in their work is regular B-Mode, but at ultra high resolution. The ultrasound technology used in their work is not available to most microsurgeons around the globe.

Using regular contemporary ultrasound devices, B-Mode in general has its domain rather in displaying tissue morphology, but does not provide any information about flow direction (value for distinguishing veins from arteries, especially in small microvessels) and flow velocity. 

 We refer to this important article in the discussion.

The novelty of our study is that it provides a structured methodology for not only for SCIP flap micro vessel detection, but also characterization applying Color-Coded Duplex Sonography (CCDS). CCDS features Color Flow (CF)-mode as well as Pulse Wave (PW) mode which are different modes from B-Mode. CF- mode is generally used to display blood flow, blood flow direction and measure vessel diameter. It is therefore ideal not only to detect but further assess and characterize blood vessels. PW-Mode adds even more information to further evaluate SCIP-Vessels as blood flow is quantified. Small arteries may be reliably distinguished from veins. In this mode, the “Pulse-Systolic-Velocity” (PS) and “End-Diastolic-Velocity” (ED) as well as the “Resistance-Index” (RI) may be measured. Vascular diameter and blood flow velocities may then serve as an orientation to compare the different perforators. Thus, different modes and a characterization of vascular anatomy for SCIP flaps are presented in our study. Reference values are provided for comparability.  Moreover, our paper describes precise device settings how to precisely trim ultra sound devices to make them applicable for the design of thin perforator flaps as the SCIP. Our paper is supposed to address microsurgeons interested in learning CCDS technology. The paper also provides a structured methodology that simplifies pre-operative markings and flap design in order to improve safety and efficiency. In comparison with Yoshimatsu’s work, the device applied in our study using a 15 MHz transducer is by far more comparable to any standard contemporary ultrasound machine available in almost any larger hospital. Thus, the target group of microsurgeons having devices available to them such as the device used in our study is by far much larger than the ones with access to ultra-high-performance devices with transducers of 48 MHz such as it was used in their study.

We therefore think there might be rather an apparent resemblance between the study of Yoshimatsu et al. and our study. However, both studies rather complement each other.

We included this important article in the discussion.

Changed parts in the manuscript are marked up using “track changes”.

Round 2

Reviewer 1 Report

Dear authors, 

Thank you for revising the manuscript and adding important aspects to the methodology, results and discussion. 

Thank you for the opportunity to review your work and best of luck in your further scientific endeavours.